

# New alignment-based sequence extraction software (ALiBaSeq) and its utility for deep level phylogenetics

Alexander Knyshov[1], Eric R.L. Gordon[2] and Christiane Weirauch[1]

[1] Department of Entomology, University of California, Riverside, Riverside, CA, USA
[2] Department of Ecology and Evolutionary Biology, University of Connecticut, Storrs, CT, USA

## ABSTRACT

Despite many bioinformatic solutions for analyzing sequencing data, few options exist for targeted sequence retrieval from whole genomic sequencing (WGS) data with the ultimate goal of generating a phylogeny. Available tools especially struggle at deep phylogenetic levels and necessitate amino-acid space searches, which may increase rates of false positive results. Many tools are also difficult to install and may lack adequate user resources. Here, we describe a program that uses freely available similarity search tools to find homologs in assembled WGS data with unparalleled freedom to modify parameters. We evaluate its performance compared to other commonly used bioinformatics tools on two divergent insect species (>200 My) for which annotated genomes exist, and on one large set each of highly conserved and more variable loci. Our software is capable of retrieving orthologs from well-curated or unannotated, low or high depth shotgun, and target capture assemblies as well or better than other software as assessed by recovering the most genes with maximal coverage and with a low rate of false positives throughout all datasets. When assessing this combination of criteria, ALiBaSeq is frequently the best evaluated tool for gathering the most comprehensive and accurate phylogenetic alignments on all types of data tested. The software (implemented in Python), tutorials, and manual are freely available at https://github.com/AlexKnyshov/alibaseq.

Corresponding author
Alexander Knyshov,
aknys001@ucr.edu

## INTRODUCTION

Phylogenetic reconstructions have traditionally used only a fraction of the sequence data of an organism's genome, but due to the widespread application of Next Generation Sequencing (NGS) to phylogenetics the quantity of data continues to increase. Phylogenomic studies have therefore heavily relied on a handful of reduced representation approaches including transcriptome sequencing (RNASeq), DNA-based reduced representation techniques, and genome skimming. RNASeq was among the early, still fairly expensive, techniques to obtain large numbers of loci that are informative for deep phylogenetic divergences. Recently, the more cost-effective sequencing of targeted genomic DNA, enriched via hybrid capture, became popular and is at the core of widely used approaches including Ultra Conserved Element (UCE) (*McCormack et al., 2012*) and

Anchored Hybrid Enrichment (*Lemmon, Emme & Lemmon, 2012*) methods.
As sequencing costs have dropped during the past decade, genome skimming (low coverage whole genome sequencing) has become a viable alternative to target enrichment, at least for taxa with relatively small (1 Gbp) genomes. This technique is less challenging with respect to sample quality, involves less complicated lab protocols and does not require expensive probe synthesis. This last point is critical for sampling phylogenetically diverse taxa because the recovery of target sequences is not bound by limitations of the probe design.

While genome skimming does confer these potential benefits, the resulting data can be difficult to parse or integrate into a phylogenetic dataset and can pose substantial problems for analysis. For example, assembled sequences may differ from deep-sequenced model taxon genomes in being much less contiguous as well as unannotated. Genome skimming data also differ from RNASeq data, most notably by the presence of untranslated highly variable regions such as introns. As opposed to typical target capture data, where targeted loci have much higher coverage than non-target ones (*Knyshov, Gordon & Weirauch, 2019*), genome skimming produces more uniform coverage across the genome (*Zhang et al., 2019*), with differences associated primarily with sequence properties such as GC content (*Barbitoff et al., 2020*). Also unlike hybrid capture methodologies, where probes are typically designed for a particular set of taxa based on a related reference taxon (*Faircloth, 2017*; *Young et al., 2016*), genome skimming can be applied to taxa with or without available reference genomes or transcriptomes. Nevertheless, hybrid capture-based bioinformatic solutions are most commonly applied to the phylogenetic analysis of genome skimming data (*Chen et al., 2018*; *Zhang et al., 2019*).

Phylogenetically-oriented hybrid capture and genomic pipelines are subdivided into two main groups of approaches. Software in the first group identifies reads of interest with the help of reference sequences and subsequently assembles this limited pool of reads (aTRAM (*Allen et al., 2015*, *2018*), HybPiper (*Johnson et al., 2016*), Assexon (*Yuan et al., 2019*), Kollector (*Kucuk et al., 2017*), and HybPhyloMaker (*Fér & Schmickl, 2018*)). The search for reads that match target regions typically makes use of read aligners (HybPiper, Kollector, HybPhyloMaker) or local similarity search algorithms on both the nucleotide and protein levels (aTRAM, HybPiper, Assexon). After reads are gathered, they are fed to an assembler, and assembled contigs are further processed. A benefit of this group of approaches is that there is no need to assemble the entire read pool, making them potentially faster and less memory demanding than approaches that use the whole read pool. Some drawbacks are the need to perform new read searches and assemblies for each new set of baits and the inability to work with assembled data.

The second group of approaches uses an assembly compiled from the total read pool. The assembly is queried for target sequences, which are then extracted and processed. Post-assembly dataset-specific target searches can be performed relatively quickly. However, especially for highly divergent taxa, the assembly process itself may be both a memory- and time-demanding procedure. Generating a set of contigs from transcriptomic assemblies can be relatively straightforward, because they mostly consist of spliced

protein coding sequences. This approach is utilized in HaMStR (*Ebersberger, Strauss & Von Haeseler, 2009*), Orthograph (*Petersen et al., 2017*), Orthofinder (*Emms & Kelly, 2019*), and FortyTwo (*Simion et al., 2017*), among other applications. However, unannotated genomic assemblies may have contigs comprised of multiple genes or untranslatable introns of varying size. Gene prediction and protein extraction may be complicated when a target gene is fragmented into many small contigs. Recently, *Zhang et al. (2019)* suggested using Phyluce (*Faircloth, 2016*) for UCE extraction and Benchmarking Using Single Copy Orthologs (BUSCO) (*Simão et al., 2015*; *Waterhouse et al., 2017*) for OrthoDB Single Copy Ortholog (SCO) extraction from genomes at shallow phylogenetic levels, that is, from relatively closely related taxa. Between these two solutions, only BUSCO is specifically designed for genomic assemblies and has the capability to search for and predict genes de novo, but it is only feasible for a few predetermined sets of proteins. Phyluce was originally designed for short, conserved fragments and it is unclear how well it performs on longer multiexon genes. The recently published Assexon software (*Yuan et al., 2019*) is capable of searching for and retrieving sequences from genomic assemblies, but this module has not yet been extensively tested.

To address issues with commonly-used techniques for including genome-skimming data in phylogenies, we have developed a software, named ALiBaSeq (ALignment Based Sequence extraction), that is designed for sequence extraction based on a local alignment search and is applicable to all types of assembled data and a wide range of assembly qualities. The software is flexible with respect to both input and output, which will facilitate its incorporation into existing bioinformatics pipelines. Any read processing technique and assembler are supported to generate the input for the software, while the resulting sequences are output in FASTA format and can be grouped in several ways (per target locus, per sample, etc.) depending on what is required in downstream analyses. The software also allows for the integration of different types of datasets (e.g., transcriptomic and sequence capture data) allowing phylogenies with more complete taxon sampling as these various phylogenomic datasets become more and more available (*Kieran et al., 2019*). One of the software's particular strengths is its ability to efficiently obtain orthologous regions from unannotated genome skimming data. Existing tools frequently rely on a particular type of sequence aligners (BLAST (*Altschul et al., 1990*) for aTRAM and FortyTwo, both BLAST and BWA (*Li & Durbin, 2009*) for HybPiper, Usearch for Assexon, LASTZ (*Harris, 2007*) for Phyluce). Our software supports several commonly utilized similarity search programs and their outputs. While we provide utility scripts for some of the tools, the aforementioned search programs can be run on their own, thus giving the user full control over search program settings if needed. Finally, compared to other programs, we offer greater customization of parameters, including different alignment score cutoff criteria, specification of number of alternative matches, and sequence output structure. The software is available for download at https://github. com/AlexKnyshov/alibaseq.

We here describe the implementation of this software, assess its performance, and benchmark it against other commonly utilized algorithms. Tests are conducted on (1) both

conserved and variable loci as determined by average pairwise sequence distance, on (2) contiguous whole genome assembly, short read assemblies of variable depth of coverage, and a hybrid capture sample. We focus testing on the insect samples (see below), but also perform a subset of tests on a plant system to verify the software's versatility, the details of which are available in the Text S1. Overall, we find that our software matches or outperforms other techniques applied to genome skimming data in recovering the most orthologous genes with the lowest amount of error in low-coverage, fragmented and unannotated genome assemblies. Furthermore, we determine that it works as well or better than other tools on high coverage genome assemblies and target capture assemblies especially at relatively deep phylogenetic levels (100–200 Mya). Thus, ALiBaSeq is a valuable tool for compilation of phylogenomic datasets across diverse taxa and diverse data types.

# MATERIALS AND METHODS

## Algorithm and implementation

The workflow is shown in Fig. 1 and the terminology is outlined in Table 1. The input alignment table can be generated by a number of programs, with supported formats including BLAST (blastn, blastp, blastx, tblastn, tblastx), 15-, 18-, and 22-field HMMER (*Eddy, 2011*) formats, Phyluce-style LASTZ output format, as well as SAM/BAM alignment formats (refer to online manual for more information). The software parses the input table and groups all HSPs for each contig (hit). For each query-hit pair, HSP hit regions are merged when overlapping and joined in the case of no overlap, producing a pseudocontig. Contig regions derived from HSPs with overlap only in the bait sequence (different contig regions matching the same bait region) are separated into alternative "pseudocontigs". This accounts for a case of a contiguous assembly where several homologous genes occupy the same chromosomal contig and could be located nearby. Thus, no preliminary contig splitting (for example, as done in Phyluce) is needed for genomic data. For the translated searches, the hits derived from the opposite strands are kept as separate "pseudocontigs". By default, each particular region of every contig is allowed to match only one bait (to prevent the same sequence from being assigned to multiple baits).

An optional reciprocal best hit (RBH) check can ensure that each pseudocontig which matches a particular bait also matches the bait-matching contig in the reference assembly. This check removes out-paralogs and is also a part of Orthograph, FortyTwo, and Assexon, although is not implemented in most popular target-enrichment bioinformatics pipelines (e.g., aTRAM, HybPiper, and Phyluce). A strict RBH check however may represent a too conservative solution, especially when a DNA-based reciprocal search was performed on divergent taxa. The algorithm may result in too many false negatives, the truly homologous contigs that did not get any significant matches to a reference taxon. Thus, we added a relaxed option of retaining the contigs without matches to the reference.

After verifying that all pseudocontigs match best to the bait they were initially assigned to, they are ranked by alignment scores. Bitscore, e-value, and identity are used for ranking

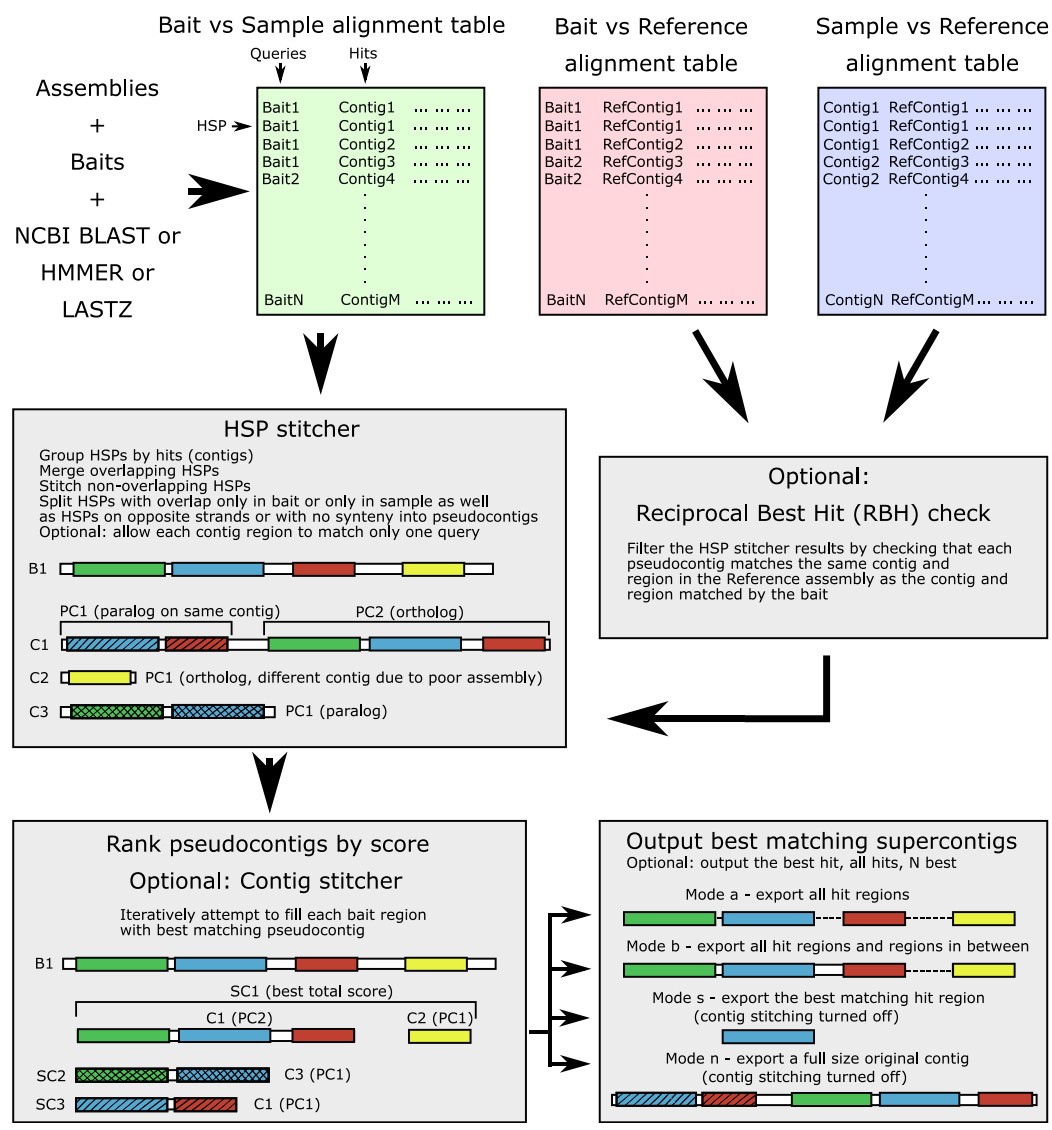

**Figure 1  Software workflow.** The workflow is illustrated by a hypothetical example of a bait for which both an ortholog and several paralogs are present in the sample. For terminology see Table 1. White boxes show contig boundaries, colored boxes represent similarity regions (shown both for the bait and the sample), shading denotes different paralogous genes. Abbreviations are as follows: B1, bait sequence for the locus 1; C1-3, assembly contigs 1-3; PC1-2, pseudocontigs 1-2; SC1-3, supercontigs 1-3; HSP, high-scoring segment pair.               

and the user has the option to specify in which order these metrics are used (in case of a tie, the next one will be used to resolve the order). We additionally provide the option (1) to use both bitscore and e-value and consider identity in case of ties; (2) to rescale the metrics by HSP length; and (3) to pick several suboptimal hits with a specifiable threshold (for more information please see the manual). Optional contig stitching is then performed by attempting to iteratively fill all regions of the bait by non-overlapping pseudocontigs. Only pseudocontigs derived from different contigs are allowed to be stitched. Stitched supercontigs receive combined scores of the included contigs to prevent

**Table 1 Terminology used in the article.**

| Term | Definition |
|---|---|
| Bait | Sequence in reference organism for which homologs are to be found in the sample |
| False positive | In context of this article, a retrieved sequence that is not matching the ortholog that should have been retrieved. Most often these are paralogous sequences |
| Forward search | Operation to match bait sequences to assembly contig sequences |
| Genome skimming | Low-depth whole genome sequencing |
| High-scoring Segment Pair (HSP) | A local alignment between the query region and the hit region |
| Hit (= subject) | Sequence with a match to query found by the local alignment search tool, a contig in a typical ALiBaSeq's application |
| Pseudocontig | Part of the contig, comprised of one or several HSPs, produced by the HSP stitcher; for the small contigs, that is transcriptomic or hybrid capture, a pseudocontig may incorporate most of the original contig, while for chromosome-sized contigs, a pseudocontig comprises a small region of the original contig |
| Query | Sequence used for search, a bait sequence in case of the forward search, and a contig sequence in case of the reciprocal search |
| Reciprocal search | Operation to match a sample assembly contig sequences to the reference assembly contig sequences |
| Supercontig | Several pseudocontigs stitched together if contig stitching is enabled; only pseudocontigs derived from different targets (original contigs) are allowed to be stitched |
| Sequence extraction/ retrieval | For the purposes of this manuscript, a procedure of searching and outputting sequence regions homologous to baits from a collection of sequences (e.g., an assembly file) |
| Target | Sequence in a sample that is homologous to the bait |

longer incorrect matches ranking higher than the composite correct match. A similar operation is done by HMMER when combining the protein domain scores into an overall hit score. While potentially allowing for chimeric sequence creation, contig stitching greatly improves coverage of long, multi-exon loci mined out of discontiguous assemblies. Apart from ALiBaSeq, a contig stitcher is also implemented in Orthograph, while aTRAM and HybPiper are among the only read-based tools to perform stitching of shorter assembled contigs.

The resulting supercontigs are then ranked by scores again, and the best supercontig or several high scoring supercontigs are then selected. We also implemented an option to select several competing supercontigs with similar scores matching the same bait in case the user wants to perform a more thorough tree-based orthology prediction. The procedure outlined thus far runs without access to an assembly FASTA file and is solely based on the alignment tables. We currently support BLAST, HMMER, LASTZ searches, as well as read aligners with SAM/BAM outputs, with a helper script enabling a tblastn-like search for HMMER.

After the list of matching sequences is prepared, the target assembly file is read, and sequences are extracted. Several extraction modes are offered: whole contigs (no contig stitching allowed), only single best matching region, all matching regions, and all matching regions plus the sequence in between (typically to get introns and variable regions as well). Additionally, extraction of flanking regions is permitted, although no similarity check is implemented for those regions. By default, the output sequences are grouped by locus and appended to existing alignments, but they may also be grouped by the sample

from which they originated, or can all be written into a single file. Output sequences can also be translated if a protein-based search is used and only matching regions are retrieved. Additionally, extensive logs and bait-sample correspondence tables with contig names, combined scores, and coordinates are saved.

The main features of the software algorithm and implementation are: assembly-based (reads are not required as input, thus all types of assemblies can be used as input, including many assemblies already available on NCBI which were based on legacy sequencing methods); lack of need for sequence annotations (local alignment search is used to determine homology and find target regions, only matching regions, for example, exons, can be output, with non-matching regions, for example, introns, not being reported); RBH check (sample contigs found are checked for match to contigs in a reference assembly from which baits were derived or to which baits are known to be homologous); contig stitching (genes broken up into multiple small contigs due to low coverage or low complexity intronic regions can be pieced together); universality (ALiBaSeq can work with inputs produced by various preceding assembly and local alignment search steps, and the output of ALiBaSeq is suitable for various subsequent multiple sequence alignment, trimming, and phylogenetic reconstruction steps).

## Samples

In order to evaluate the software's performance, namely amount of data retrieved, proportion of false positives, and speed, at deep phylogenetic levels, we used the two species *Rhodnius prolixus* and *Cimex lectularius*. They belong to the two distantly related families, Reduviidae and Cimicidae, respectively, within the heteropteran infraorder Cimicomorpha (Hemiptera). The divergence between the two species was recently dated to the mid-Triassic, roughly 225 million years ago (*Johnson et al., 2018*). Since genomes and annotations are available for both (*Mesquita et al., 2015*; *Rosenfeld et al., 2016*), we tested the retrieval of *Rhodnius* sequences using divergent *Cimex* bait sequences for orthologs known to be shared by both species.

We downloaded the original assemblies of the *Cimex* and *Rhodnius* genomes (ClecH1 and RproC3) and transcripts/gene sets (ClecH1.3 and RproC3.3) from VectorBase (*Giraldo-Calderón et al., 2015*). To test and compare both read-based and assembly-based software, we simulated Illumina reads derived from the *Rhodnius* assembly using Art (*Huang et al., 2011*). Six simulations at various depths (1×, 3×, 5×, 10×, 20×, 40×) were performed, with uniform coverage across target and off-target regions. We used recently published results (*Kieran et al., 2019*) of empirical enrichment of a closely related species, *Rhodnius robustus*, as a target capture sample for the UCE dataset (see below). To review assembly-based software equally, we processed all reads in a similar way, that is, with Clumpify (*Bushnell, 2014*) and Trimmomatic (*Bolger, Lohse & Usadel, 2014*) and assembled them using SPAdes (*Bankevich et al., 2012*). Properties of all used assemblies were evaluated using QUAST (*Gurevich et al., 2013*) and are shown in Table 2. Whenever non-default settings were used, they were indicated in Table S2.

**Table 2 Samples used for software evaluation in conjunction with UCE and ODB SCO datasets.**

| Sample | Source and version/accession | Number of PE clusters | Assembly Size (bp) | Assembly N contigs | Assembly N50 | Assembly L50 |
|---|---|---|---|---|---|---|
| Original assembly | *R. prolixus* RproC3.3 | NA | 706,824,083 | 16,537 | 1,088,772 | 170 |
| Art 1x | *R. prolixus* RproC3.3 | 1,853,060 | 55,221,538 | 103,921 | 524 | 35,823 |
| Art 3x | *R. prolixus* RproC3.3 | 5,548,045 | 242,206,429 | 311,385 | 923 | 84,342 |
| Art 5x | *R. prolixus* RproC3.3 | 9,244,580 | 367,224,880 | 287,779 | 1,736 | 66,353 |
| Art 10x | *R. prolixus* RproC3.3 | 18,478,995 | 580,423,518 | 1,305,726 | 3,209 | 41,579 |
| Art 20x | *R. prolixus* RproC3.3 | 36,965,240 | 539,547,878 | 740,413 | 6,029 | 20,520 |
| Art 40x | *R. prolixus* RproC3.3 | 73,929,196 | 539,002,827 | 698,488 | 6,826 | 18,378 |
| Capture | *R. robustus* SRR7819296 | 1,776,377 | 2,357,426 | 2,384 | 1,107 | 851 |

## Baits

To ensure our software was applicable for two commonly-used data types, two sets of loci were used as baits, the Ultra Conserved Elements (UCE) Hemiptera 2.7K v1 set (*Faircloth, 2017*) and the OrthoDB single copy orthologs for Hemiptera (ODB SCO). UCE loci and ODB SCO orthogroups used are listed in Text S3.

For UCEs, we extracted loci shared between *Cimex* and *Rhodnius* from the Hemiptera UCE 2.7K v1 set (*Faircloth, 2017*) and merged probes to obtain the overall bait regions (since typically two probes tile a 160 bp region). Since 99.9% of the kit was shown to target exons of protein coding genes in the taxon with the most well-annotated genome (*Kieran et al., 2019*), we prepared DNA and AA sequences for both nucleotide and protein modes of the programs we were assessing. A few particularities of the probe set caused us to perform additional filtering and processing: (1) some introns flanking UCE exons are part of the probes despite their relative high rate of mutation; (2) several UCE "loci" in *Cimex* and/or *Rhodnius* are overlapping for about a third of their length, with probes targeting opposite strands (likely due to intron drop out in either of the taxa compared to the other probe set taxa); (3) the bait region borders (probe start/stop coordinates) are not always homologous between *Cimex* and *Rhodnius* (because of issue 1); (4) some UCE loci are too close to each other on the scaffold to exclude the possibility of capturing both loci with probes derived from only one (we did not address this issue, since it did not impede our evaluation analyses). Thus, to obtain homologous protein coding regions of the probe set, we first overlapped *Cimex* probes of the same region and aligned the resulting sequences using BLAST to *Cimex* genomic scaffolds. We randomly removed one of the UCE loci which were overlapping since these duplicate UCEs could be flagged by some of the software we were testing. We compared the mapping coordinates with the assembly annotation and truncated the probe regions to stay within the CDS. We then compared the CDS with the *Cimex* proteome to find the start and stop of the ORF and get the protein sequence of the probes. Using protein BLAST, we aligned *Cimex* AA probes to *Rhodnius* DNA probes and truncated the latter to make sure that homologous *Rhodnius* regions are used for evaluations. The resulting UCE dataset had 2,037 loci, with locus

lengths ranging from 60 to 160 bp, nucleotide pairwise sequence distances (n-psd) ranging from 1.59% to 50.69% (average 22.44%) and 0% to 60.42% (average 6.88%) on the protein level.

For ODB SCO, we retrieved orthologs that are single copy for Hemiptera from OrthoDB v9. We then extracted *Rhodnius* and *Cimex* protein sequences, aligned them with MAFFT (*Katoh & Standley, 2013*), and trimmed off lateral gapped regions. We subsetted the alignments to retain only those with >100 AA and less than 10% gaps between the two species, then used the resulting truncated protein sequences to obtain the corresponding CDS from the VectorBase sourced transcriptomes of the two species. We obtained 1,027 loci and initially used those for benchmarking. However, when investigating false positive results, we discovered that some results seemed to be correct based on the pairwise distance and were in fact correct according to the VectorBase orthology predictions. We thus compared orthology assignment of 1,027 selected loci between OrthoDB and VectorBase. We found cases where one (primarily *Rhodnius*-sourced) gene was assigned to an incorrect orthogroup on OrthoDB, as well as cases where a gene was not single-copy (had paralogs in *Cimex* and/or *Rhodnius*) according to VectorBase, while being listed as single-copy on OrthoDB. A total of 43 such loci were found and discarded from the evaluations. We additionally inspected the *Rhodnius-Cimex* alignments and discarded 10 other loci that had poor alignment between the two species, suspecting them of being not true orthologs. The resulting ODB SCO dataset had a total of 974 loci, with locus lengths ranging from 216 to 20,862 bp, and nucleotide pairwise sequence distances ranging from 15.22% to 52.89% (average 29.84%) and 0% to 61.62% (average 20.15%) on the protein level.

## Software and settings

For read-based approaches, we tested four software packages, Assexon, aTRAM, HybPiper, and Kollector (Table 3). In order to make the comparison equal, we used processed reads for the SPAdes assemblies (see above) as an input for the read-based software. The default settings of the evaluated software were drastically different, and in some cases inappropriate for the deep level divergence we were looking at, causing suboptimal performance. We thus attempted to standardize them and set parameters of the programs to a very generous minimal level (Table S2). Due to specific requirements of Assexon on NGS read format, we could not run it on our simulated reads. aTRAM's performance with the SPAdes assembler was poor due to coverage cut-off values and timeout problems; when the time parameter was increased, the analyses took days to complete; thus we ran the protein-based analyses with Velvet (*Zerbino & Birney, 2008*) for evaluation purposes. HybPiper's protein-based search speed on the ODB dataset, particularly when using $20\times$ and $40\times$ reads, was extremely slow (did not complete in 8 days), thus we set the filtering settings to default for these trials to maximize locus recovery. We note that changing the HybPiper settings to default did not change the results of the low depth trials (1x–5x), while improving the results on the 10x dataset.

For assembly-based approaches, we tested Assexon, FortyTwo, Phyluce, and our software ALiBaSeq (Table 3). We note that Orthograph is very close in capabilities to

**Table 3  Software used for evaluation.**

| Software | Application | Sequence type (>> - in, << - out) | Search engine | Dependencies | Reciprocal best hit check | Reference |
|---|---|---|---|---|---|---|
| ALiBaSeq | Assembly-based | >> DNA/AA << DNA/AA | External | Python 2 or Python 3, Biopython, search tool (BLAST, HMMER, LASTZ, read aligners) | Optional | This study |
| Phyluce | Assembly-based | >> DNA << DNA | LASTZ | Python2, several python packages, LASTZ | No | *Faircloth (2016)* |
| FortyTwo | Assembly-based | >> DNA << AA | BLAST | Perl, Perl Bio::MUST, BLAST, exonerate | Optional | *Simion et al. (2017)* |
| Assexon | Read- and assembly-based | >> DNA + AA << DNA/AA | Usearch | Perl, several perl modules, Usearch, exonerate, SGA | Mandatory | *Yuan et al. (2019)* |
| aTRAM | Read-based | >> DNA/AA << DNA | BLAST | Python3, BLAST, SPAdes, exonerate | No | *Allen et al. (2018)* |
| HybPiper | Read-based | >> DNA/AA << DNA/AA | BWA, BLAST | Python2, BLAST/BWA, SPAdes, parallel, BWA, samtools | No | *Johnson et al. (2016)* |
| Kollector | Read-based | >> DNA << DNA | BWA | ABYSS, gmap, BWA, samtools | No | *Kucuk et al. (2017)* |

FortyTwo, with both programs being primarily designed to work with transcripts. However, Orthograph has a substantial list of dependencies, and requires a relational database, thus, we have not included it in our evaluation. Despite the program BUSCO's capability to process genomic data and retrieve target sequences with high precision, this software is hard to customize. The lowest level relevant gene set available are orthologs at the level of Insecta. The presence of *Rhodnius* in the profiles would not allow us to fully test the performance with a divergent reference. We thus did not include BUSCO in our comparison. Settings used are listed in the Table S2. For ALiBaSeq we ran the software both with strict and a relaxed RBH check (see implementation). Figures represent the best performing combination of such analyses (strict for 1-3x assemblies, relaxed for others), for details see Tables S3 and S4. For Phyluce, we generally used guidelines in tutorial 3 (https://phyluce.readthedocs.io/en/latest/tutorial-three.html). We had to use the 0 bp flank option to maximize the locus recovery since many loci were being discarded due to co-location on the same contig. The capture sample was processed directly starting with the SPAdes assembly, bypassing the typical read processing and Trinity assembly for comparison purposes. The phyluce_assembly_match_contigs_to_probes command for the capture sample was run with the same settings as for genomic samples.

All programs were run on the UCR High Performance Compute Center cluster. We capped all read- and assembly-based programs to run maximally with 32 threads to make the performance comparison fair. To assess accuracy of the recovery, obtained DNA (or protein, in the case of FortyTwo) sequences were aligned back to the *Rhodnius* sequences that were homologous to the *Cimex* bait sequences. The alignment was done with MAFFT G-INS-i or E-INS-i (*Katoh & Standley, 2013*); and the—adjustdirectionaccurately option was used for outputs of Kollector, since Kollector does
not maintain hit sequence direction with respect to the query sequence. Since only ALiBaSeq has the capability to report the source contig names and coordinates for the retrieved sequences, we had to rely on sequence similarity to determine whether an ortholog or a paralog was recovered. We observed that sequence similarity of less than 100% was too strict a criterion for assessing false positive results. Small alterations in simulated reads, assembly of such reads, incorrect flanking bases of the exon-intron junctions, and alignment errors lower the apparent sequence similarity when the retrieved sequence is compared to the original sequence. Thus, we used a threshold of 10% difference to track strongly false positive results. Since our empirical UCE capture sample belonged to a different species (*R. robustus*), we used a larger threshold of 20% to account for additional interspecific variation.

We also evaluated software performance for difficult loci, which either have multiple introns, or are represented by multiple contigs in the assembly (for assembly-based tools only). The evaluation of the performance on multi-intron and multi-contig loci was evaluated on the 40x ODB dataset. To assess the impact of introns on the software performance, we queried the annotation file for *Rhodnius* to obtain the number of introns per SCO, and then regressed the SCO coverage produced by each software on the number of introns using a linear model. To evaluate the impact of SCO fragmentation in the assembly, we performed a blastn search for *Rhodnius* bait homologs in the *Rhodnius* 40x read assembly, processed the results with ALiBaSeq to estimate the number of contigs representing each SCO locus, and then regressed the SCO coverage produced by each software on the number of contigs.

### Additional evaluation methods

Methods of the software evaluation using the plant dataset are available in the Text S1.

## RESULTS

### The UCE dataset

#### Number of loci recovered (Fig. 2A)

On the original assembly, ALiBaSeq recovered up to 2,032 out of 2,037 UCE loci. Protein based approaches (tblastn, tblastx, and phmmer) performed best, with relaxed nucleotide-based searches (blastn with wc9 and discontinuous megablast) performing almost as well. Both Assexon and Phyluce recovered 1,436 and 1,577 loci respectively, while FortyTwo only found ~300 loci. On the assemblies of simulated reads, ALiBaSeq generally performed best, or as well as other assembly-based software. ALiBaSeq, Phyluce, and FortyTwo reached their maximum performance at assemblies of 10x and deeper and also performed well on the empirical target capture data. Read-based methods using nucleotide space searches performed very poorly, with the exception of Assexon (~1,500 out of 2,037 loci), which we could only test on the capture sample. Protein-based aTRAM performed the best of these methods in finding the maximum number of loci on the simulated genomic reads at 5x coverage and less. Protein-based HybPiper only performed well on 40x and target capture data, with only ~1,300 out of 2,037 loci on the

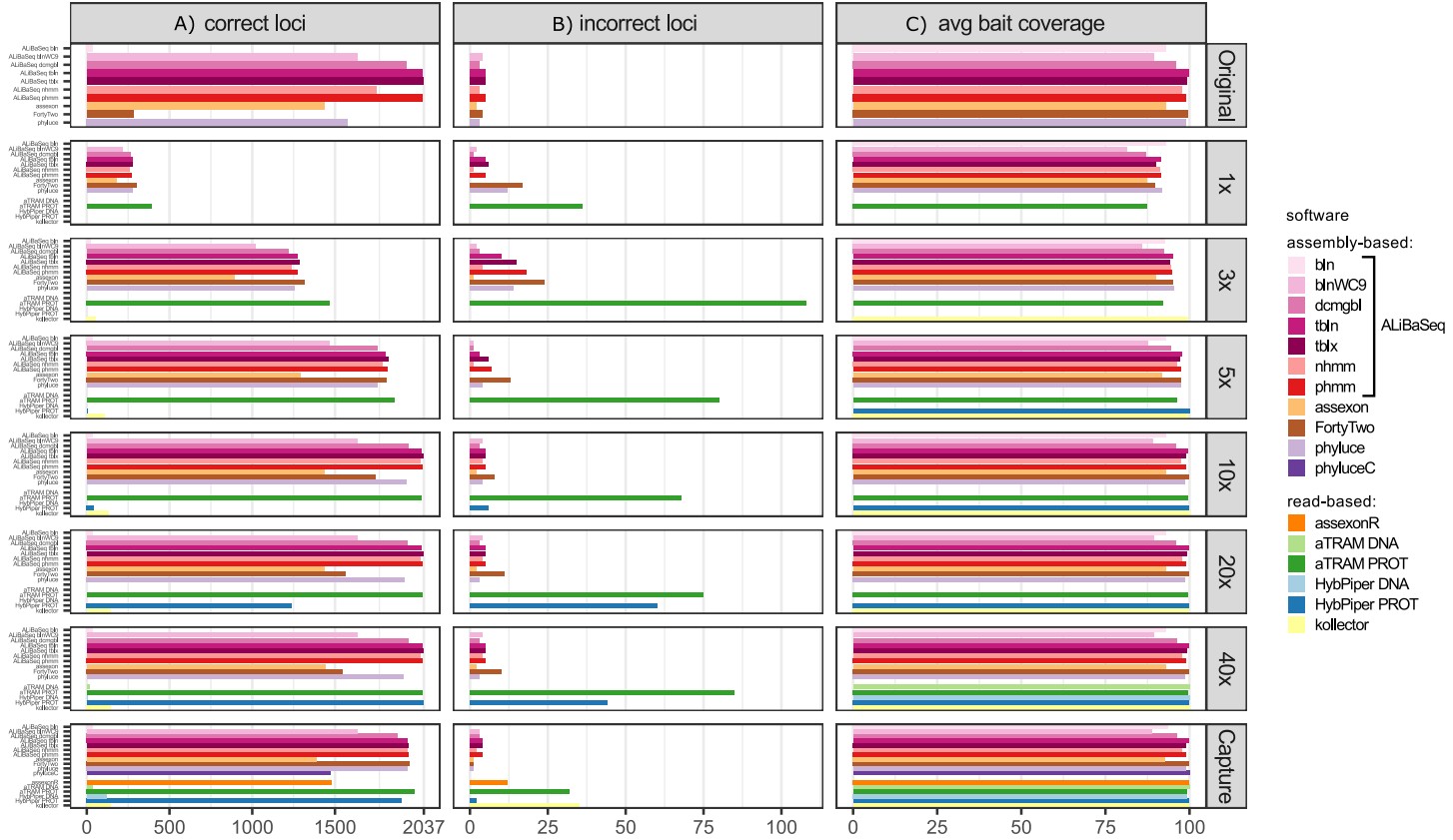

**Figure 2** **Performance on the UCE dataset.** (A) Amount of loci with a high sequence identity to the *R. prolixus* bait region sequence. (B) Amount of loci with a low sequence identity to the *R. prolixus* bait region sequence. (C) The average percentage of the bait region recovered. Vertical panels refer to different datasets (see text for details). Abbreviations are as follows: bln, blastn; blnWC9, blastn with word length 9 bp; dcmgbl, discontinuous megablast; tbln, tblastn; tblx, tblastx; nhmm, nhmmer; phmm, phmmer; phyluceC, capture pipeline of phyluce; assexonR, read-based version of Assexon.

20x data, and much less on the shallower data. Interestingly, the genomic pipeline of Phyluce performed better (1,939 loci) on the target capture data of *Kieran et al. (2019)* than the standard Phyluce capture pipeline (1,471 loci), a result closer to that reported in the original study (*Kieran et al., 2019*) for this sample (1,508 loci).

### *False positives (Fig. 2B)*

All tools performed well on the original assembly (0.0–0.25% false positives) with the exception of FortyTwo (1.4%). The DNA-based ALiBaSeq search on simulated read assemblies had less than 1% false positives. Protein-based ALiBaSeq had a higher proportion of false positives (1–2%) on the low-coverage assemblies (1x and 3x respectively), with a much lower proportion (0.25%) for 10x–40x assemblies. This is comparable to other assembly-based software false positive rates, and much lower than that of read-based programs: up to 9% for protein-based aTRAM and up to 12% for protein-based HybPiper. Lower-depth (1–3x) reads and read-based assemblies were more challenging for accurate orthology prediction. The conservative approach of exclusion of contigs that lacked any hits in the reciprocal table improved orthology

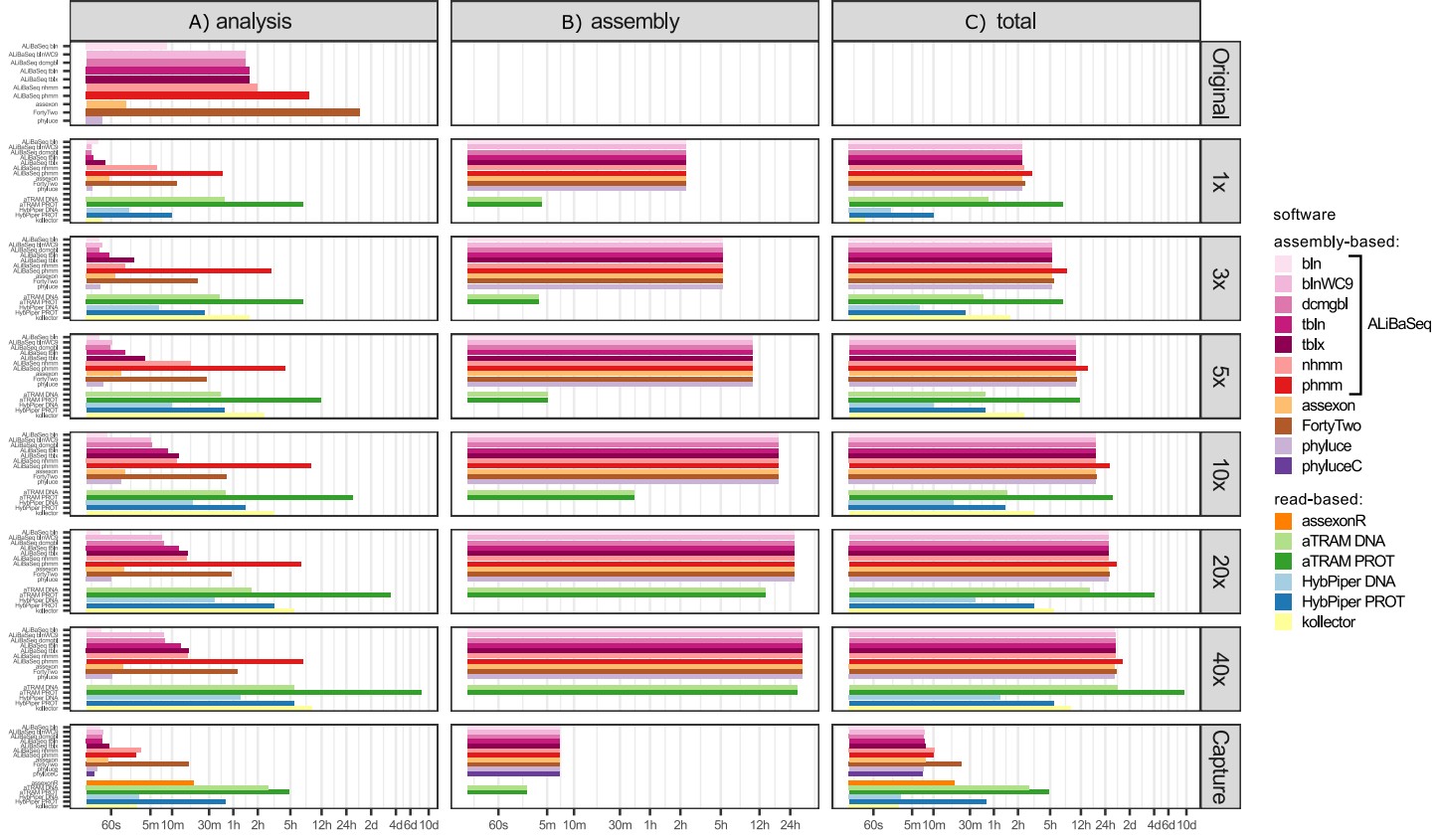

**Figure 3** **Speed of the processing of the UCE dataset.** (A) The search and sequence retrieval time (including assembly in case of the read-based tools). (B) The assembly stage time for the read-based tools or the database creation time for aTRAM. (C) The total time elapsed. *X* axis scale is log-transformed.                                             

assignment for 1–5x data however worsened the results of 10–40x data analysis (true orthologs were incorrectly discarded, see Table S3).

### Bait region coverage (Fig. 2C)
As UCE bait regions are short, generally ALiBaSeq and other tools recovered almost complete bait regions (~80–100% of the length), with a minimum of 78% recovered on the lowest coverage assembly.

### Time (Fig. 3)
Depending on the search type, the ALiBaSeq pipeline on the original assembly completed a forward BLAST search in between 2 s (dc-megablast) and 4 min (tblastx) and a reciprocal search in between 8 and 90 min. HMMER searches ran much longer (up to 7 h). The main script ran for 15 s to 2 min. The total processing time for ALiBaSeq thus was between 8 min and 1.5 h with BLAST searches, and up to 8 h with HMMER searches. Assexon and Phyluce worked faster than ALiBaSeq searches (30 s–2 min total time), while FortyTwo was slower (several days). The same relationship was generally true for the searches on simulated read assemblies. On smaller datasets (1–5x), ALiBaSeq with nucleotide blast was as fast as Phyluce and about as fast as Assexon with protein blast.

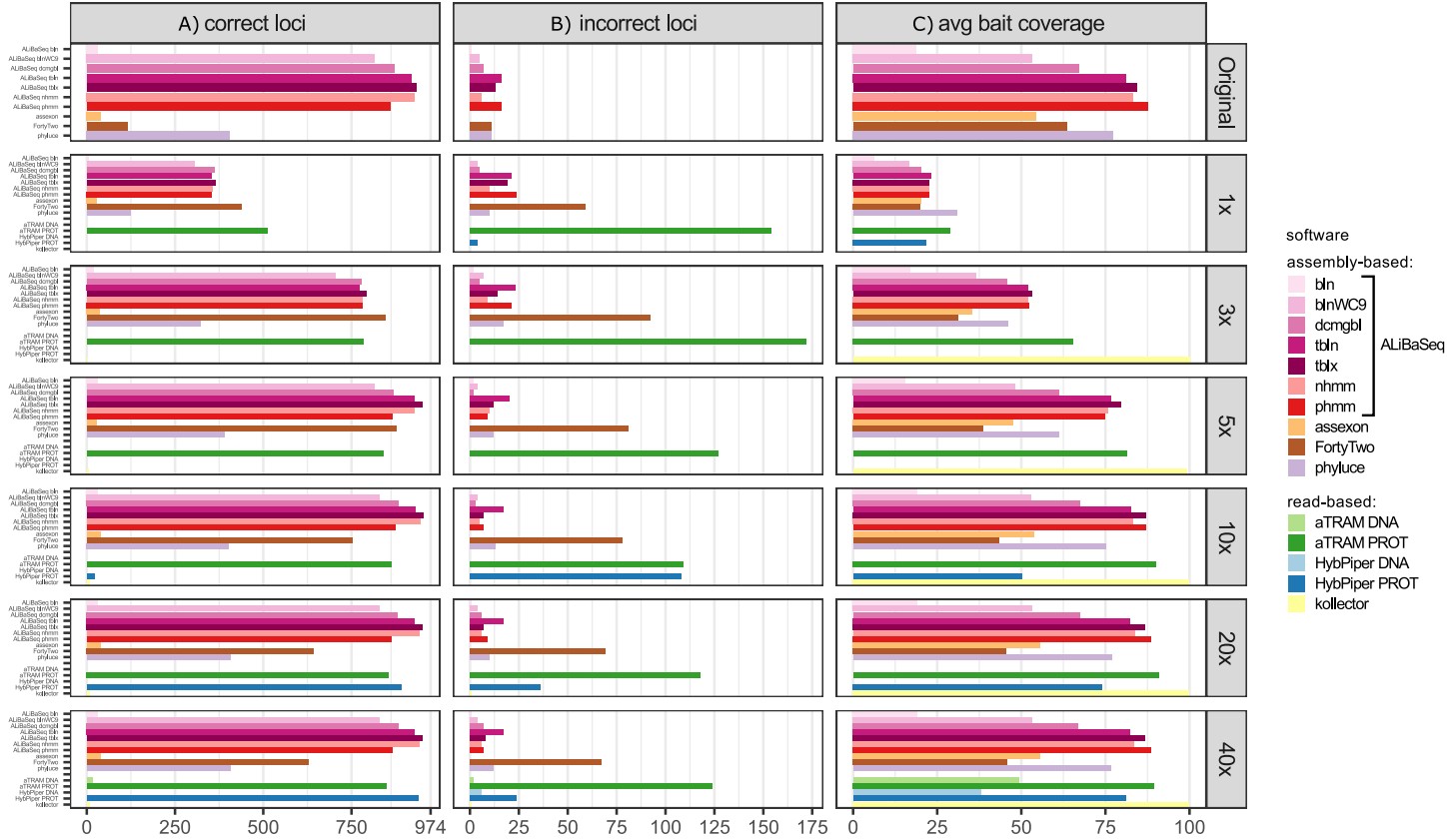

**Figure 4 Performance on the ODB SCO dataset.** (A) Amount of loci with a high sequence identity to the *R. prolixus* bait region sequence. (B) Amount of loci with a low sequence identity to the *R. prolixus* bait region sequence. (C) The average percentage of the bait region recovered. Vertical panels refer to different datasets (see text for details). Abbreviations are as follows: bln, blastn; blnWC9, blastn with word length 9 bp; dcmgbl, discontinuous megablast; tbln, tblastn; tblx, tblastx; nhmm, nhmmer; phmm, phmmer.     

DNA-based read-based tools worked very quickly, but that can be explained partially by much lower locus recovery. Adding the assembly time to the analysis time, assembly-based tools' speed (1–1.5 days) is higher on the larger datasets than that of protein-based aTRAM (20x and 40x with 4–10 days). The processing time of protein-based HybPiper on the UCE dataset (9 min–5.5 h) is smaller than the total time of assembly-based tools (2.25 h minimum). Interestingly, the SPAdes assembly time of the capture sample was much lower (~7 min) despite the comparable read number with 1x whole genome data, likely due to lower K-mer diversity. This gave the assembly-based tools a clear advantage in speed (7–17 min total time) over the read-based tools that found a considerable amount of loci (Assexon, protein-based HybPiper, and aTRAM with 19 min, 47 min, and several days respectively).

## The ODB SCO dataset
### *Number of loci (Fig. 4A)*
ALiBaSeq found the highest number of loci (up to 933 out of 974) on the original assembly by a very large margin, twice as many as the best alternative (404 out of 974 for Phyluce). On the assemblies of simulated reads, FortyTwo found most loci in the low coverage

assemblies (1–3x). ALiBaSeq performed well on low coverage assemblies and the best of all assembly-based tools on assemblies of 10x and deeper. As in the UCE dataset, only protein-based versions of the read-based tools found any significant number of loci. The software aTRAM outperformed HybPiper on low coverage data (1–10x) but recovered fewer loci than HybPiper when analyzing 20x and 40x reads.

### False positives (Fig. 4B)

The highest levels of false positives were found in the results of the read-based tools: protein-based aTRAM with 11–23% and protein-based HybPiper with 3–100%. Among assembly-based tools, ALiBaSeq had a rate of 1–3% on nucleotide and protein levels on complete assemblies, but up to 6.4% on protein-level on 1x data. This rate was comparable or lower among the programs finding >100 loci, with Phyluce having 8% and FortyTwo 12% false positives. Most of ALiBaSeq false-positives are due to non-orthologous reciprocal best hit results (since we conducted the reciprocal searches on the nucleotide level) as well as paralogous longer hits having a much larger score than the sum of small orthologous hits (i.e., paralogs with less introns were preferred over orthologs with more introns).

### Bait region coverage (Fig. 4C)

Protein-based aTRAM and Kollector recovered the largest regions (>85%), followed by ALiBaSeq and Phyluce. Since ODB SCO baits were much larger than UCE baits, much smaller portions of the bait regions (up to 29%) were recovered from the lower coverage assemblies (1x–5x), with recovery reaching maximum values on 10x and deeper data.

### Time (Fig. 5)

Processing time on this dataset was longer compared to the UCE dataset, likely due to the larger size of the loci. However, relative performance of the tools was generally similar between the two datasets. The notable exception was HybPiper, with its exonerate (*Slater & Birney, 2005*) steps taking much longer on 20x and 40x data leading to a slower performance compared to aTRAM on the same data.

### Target-adjacent sequences

Since the majority of software did not output non-target sequences with the settings used in our analyses, we did not perform a formal evaluation of all tools. Below we outline detailed features of the programs pertinent to target-adjacent sequence extraction.

### External non-target sequences (flanks)

Flanks are retrieved by HybPiper and Kollector and are bounded by the read size. Although ALiBaSeq has the capability to extract flanks, we have not used it for the evaluation analyses. Phyluce is capable of extracting flanks, however it discards loci which overlap in flanking regions, thus we set flanks to 0bp during the evaluations to maximize locus recovery. The output of the aTRAM main script has flanks bounded by the read size, however in order to maximize locus coverage we had to run the exon stitching pipeline, which outputs a spliced sequence devoid of flanks. Similarly, Assexon and FortyTwo output spliced sequences without flanks.
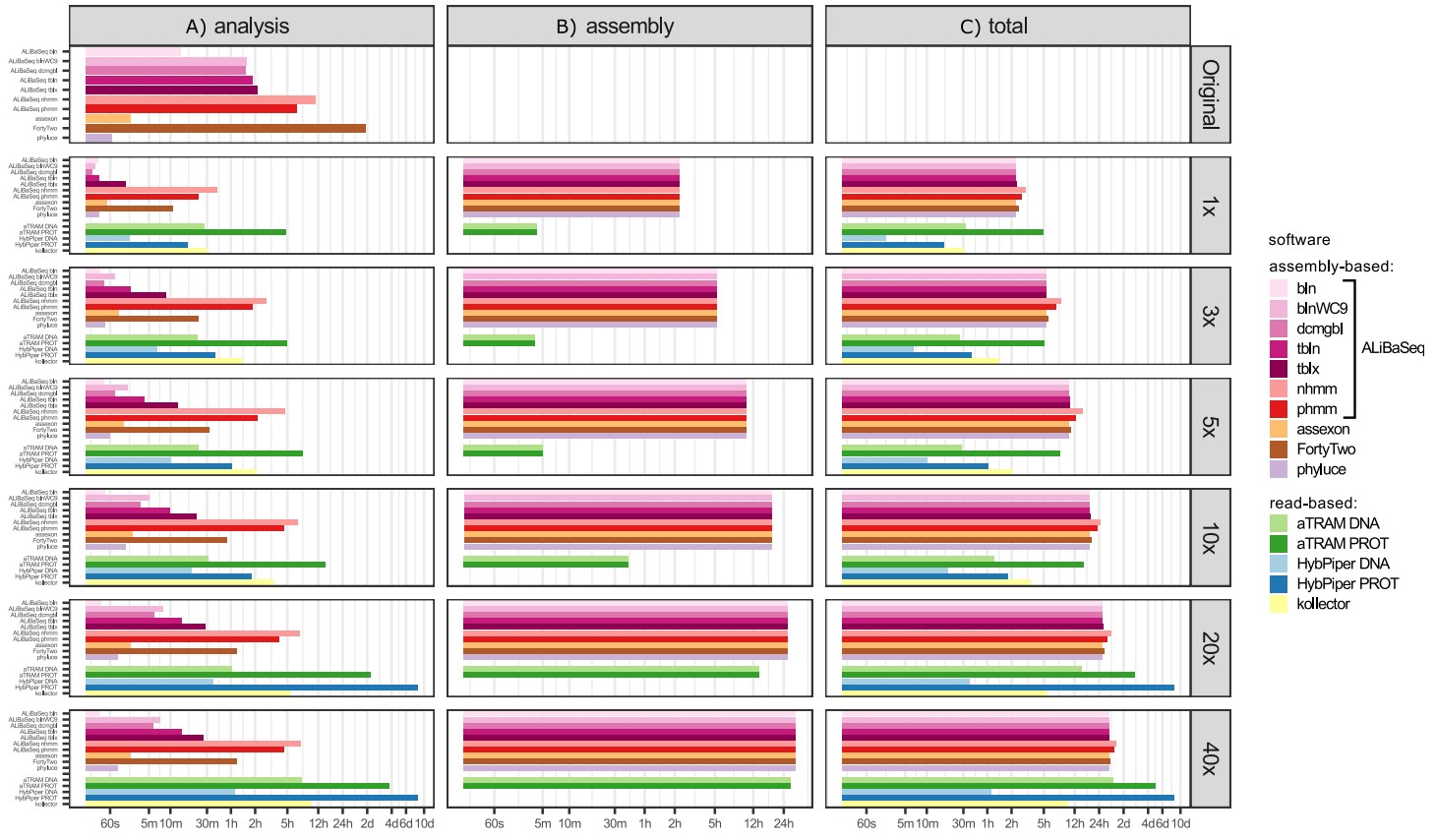

**Figure 5 Speed of the processing of the ODB SCO dataset.** (A) The search and sequence retrieval time (including assembly in case of the read-based tools). (B) The assembly stage time for the read-based tools or the database creation time for aTRAM. (C) The total time elapsed. *X* axis scale is log-transformed.                           

### Internal non-target sequences (introns)

HybPiper, Kollector and Phyluce extracted introns as part of their output in our analyses, with introns of the two former programs often being partial and bound by the read size. ALiBaSeq has a capability to extract internal non-target regions, while Assexon, aTRAM, and FortyTwo output a spliced sequence without introns.

### Number of introns vs coverage (*Fig. 6*)

ALiBaSeq in conjunction with HMMER and protein-based BLAST searches retrieved the largest regions of the multi-intron genes among all other assembly-based software. Phyluce and Assexon generally failed to recover genes with over 10 introns ($N = 38$), while FortyTwo retrieved less than 25% of the sequence of the genes with over 10 introns. Among the read-based software, protein-based aTRAM and HybPiper recovered the largest regions of the multi-intron genes, slightly outperforming ALiBaSeq.

### Number of contigs vs coverage (*Fig. 7*)

Among the assembly-based software only ALiBaSeq performed well in finding and stitching contigs comprising multi-contig genes. Assexon barely reported any multi-contig

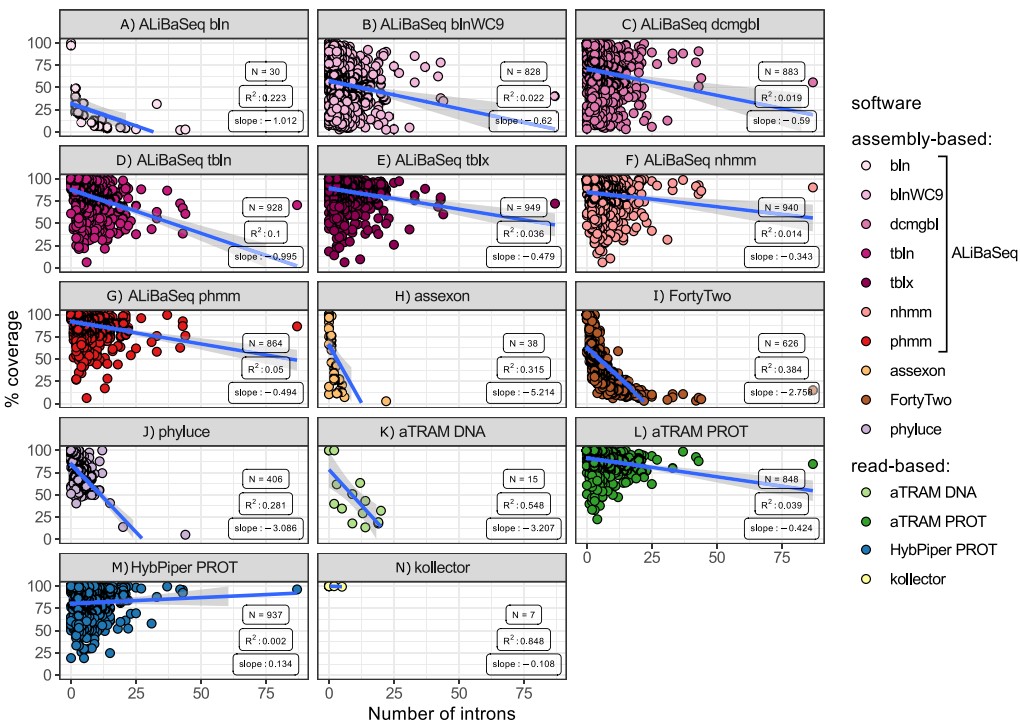

**Figure 6 Relationship between the number of introns in a locus and its coverage in the software output.** Relationship between the number of introns in a locus and its coverage in the software output. Points represent correct loci found by each program using the 40x ODB SCO dataset. (A) ALiBaSeq with blastn. (B) ALiBaSeq with blastn with word length 9 bp. (C) ALiBaSeq with discontinuous megablast. (D) ALiBaSeq with tblastn. (E) ALiBaSeq with tblastx. (F) ALiBaSeq with nhmmer. (G) ALiBaSeq with phmmer. (H) assexon. (I) FortyTwo. (J) phyluce. (K) Nucleotide-based aTRAM. (L) Protein-based aTRAM. (M) Protein-based HybPiper. (N) kollector.

genes, while FortyTwo and Phyluce output highly incomplete sequences for loci represented by three or more contigs in the assembly.

## Additional results

Results of the software evaluation using the plant dataset are available in the Text S1.

## DISCUSSION

We show that ALiBaSeq is a versatile tool, capable of retrieving orthologs from curated contiguous assemblies, low and high depth shotgun assemblies, and target capture data (Table 4). ALiBaSeq was able to accurately detect orthologs without prior annotation of the assemblies and recovered much longer regions than other assembly-based tools in part due to stitching the contigs representing different parts of the same gene in low coverage assemblies (Table 4). While utilizing a divergent reference taxon, separated from the sample by over 200 My of evolution, ALiBaSeq performed as well or better than other software in finding most genes with maximal coverage breadth and has a comparable rate of false positives throughout all datasets. With respect to the most similar programs, Assexon and FortyTwo, our program recovered more and longer sequences compared to the former, and was much faster in recovering longer sequences with a lower false positive

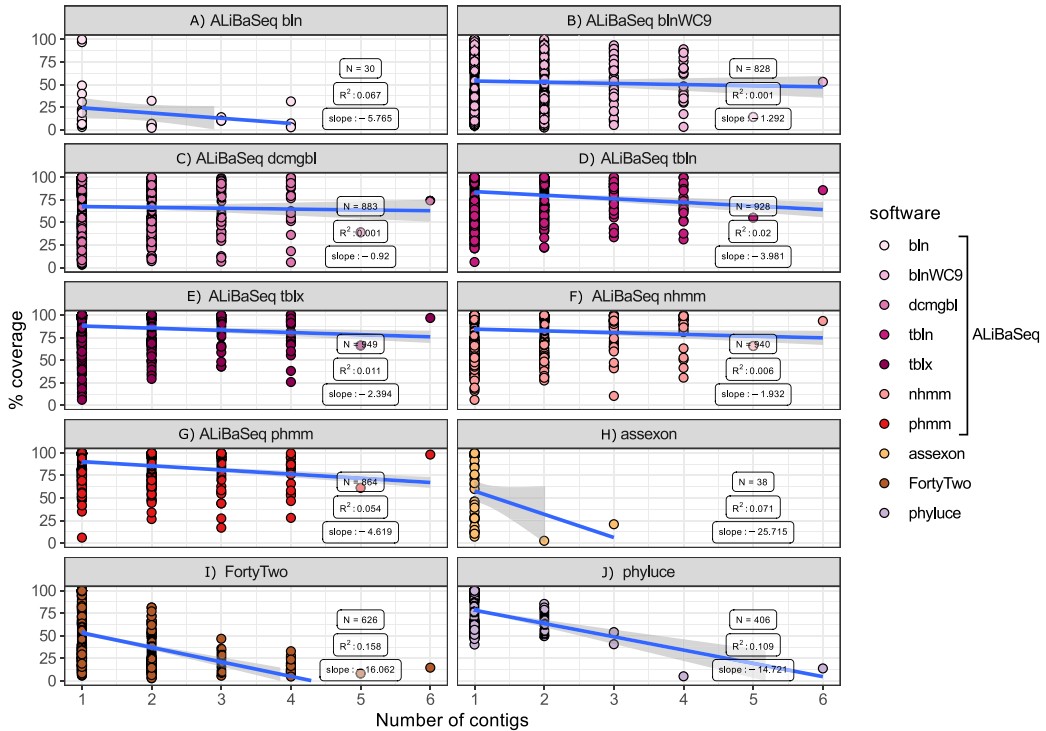

**Figure 7 Relationship between the number of contigs that a locus is represented by in the assembly and its coverage in the software output.** Only assembly-based programs are shown. Points represent correct loci found by each program using the 40x ODB SCO dataset. (A) ALiBaSeq with blastn. (B) ALiBaSeq with blastn with word length 9 bp. (C) ALiBaSeq with discontinuous megablast. (D) ALiBaSeq with tblastn. (E) ALiBaSeq with tblastx. (F) ALiBaSeq with nhmmer. (G) ALiBaSeq with phmmer. (H) assexon. (I) FortyTwo. (J) phyluce.

rate compared to the latter as well as outperforming it in correct locus recovery at higher coverage assemblies. We highlight that this outcome applies to deep level divergences between the bait taxon and the sample taxon. This conclusion will likely not be valid for shallow-level datasets, since all tested programs perform well in retrieving highly similar sequences in our evaluations, and some can do it even faster than ALiBaSeq (e.g., Assexon and Phyluce). With respect to accuracy at the protein level, additional improvements may be achieved by running the protein-based reciprocal search instead of the nucleotide-based as we did in our analyses, as well as removing outliers using alignment and gene tree screening software (*De Vienne, Ollier & Aguileta, 2012*; *Kocot et al., 2013*; *Borowiec, 2019*).

Performance tests demonstrated (Fig. 3) that for deeply sequenced samples (10x and above), whole read pool assembly followed by target searches can be completed much faster than analyses using read-based tools. At the depths where read-based tools run faster, they are unable to retrieve the same number of loci. We emphasize that this outcome is a consequence of protein-based searches for deep level analyses, since much faster DNA-based search and read aligners can be successfully used in shallow-level applications. In addition, read-based tools have a higher proportion of false-positives compared to assembly-based methods, most likely due to the lack of RBH checks. With regard to

**Table 4 Performance assessment summary.** Bold entries highlight the best performing method in a given category where applicable.

| | ALiBaSeq (tblastx) | Assexon | FortyTwo | Phyluce | aTRAM (protein) | HybPiper (protein) | Kollector |
|---|---|---|---|---|---|---|---|
| Intended input sequence/data type | Assemblies, any type | Reads and assemblies, any type | Assemblies, transcriptomic | Reads and assemblies, any type | Reads, genomic | Reads, hybrid capture | Reads, genomic |
| Possible bait to target divergence | Low to high | Low | Low to high | Low | Low to high | Low to high | Low |
| Amount of data obtained on UCE | **14–99% (avg 83%)** | 9–71% (avg 58%) | 14–98% (avg 65%) | 14–95% (avg 77%) | 19–100% (avg 83%) | 0–100% (avg 37%) | 0–7% (avg 5%) |
| Amount of data obtained on ODB SCO | **37–98% (avg 86%)** | 3–4% (avg 4%) | 12–90% (avg 63%) | 13–42% (avg 36%) | 53–89% (avg 80%) | 0–96% (avg 32%) | 0–1% (avg 0%) |
| Amount of false positives on UCE | 0.2–2.1% (avg 0.6%) | **0–0.1% (avg 0.1%)** | 0.1–5.3% (avg 1.4%) | 0.1–4.2% (avg 0.8%) | 1.6–8.4% (avg 4.6%) | 0–12% (avg 3.8%) | 0–19.9% (avg 3.3%) |
| Amount of false positives on ODB SCO | 0.7–5.0% (avg 1.7%) | **0% (avg 0%)** | 8.5–11.9% (avg 9.7%) | 2.4–7.5% (avg 3.8%) | 11.2–23.1% (avg 15.1%) | 2.5–100% (avg 47.5%) | 0–12.5% (avg 4.7%) |
| UCE locus completeness | 90–99% (avg 97%) | 87–93% (avg 92%) | **90–100% (avg 98%)** | 92–99% (avg 97%) | 87–100% (avg 96%) | 0–100% (avg 71%) | 0–100% (avg 86%) |
| ODB SCO locus completeness | 23–87% (avg 72%) | 20–56% (avg 46%) | 20–64% (avg 41%) | 31–77% (avg 64%) | 29–91% (avg 74%) | 0–81% (38%) | 0–100% (avg 83%) |
| Speed (assembly-based only) | 2 min–2 h | 1 min–2.5 min | 10 min–2 days | **30 s–1 min** | NA | NA | NA |
| Splicing | Optional | Yes | Yes | No | Optional but coupled with exon stitching | Optional | No |
| Performance on multi-intron genes (>20 introns) | Good | Poor | Poor | Poor | Good | Good | Poor |
| Performance on multi-contig genes (≥2 contigs) | Good | Poor | Average | Average | NA | NA | NA |

ALiBaSeq's RBH check, the relaxed option of not discarding the contigs that did not have any hits in the reciprocal tables proved beneficial for the deep coverage assemblies, resulting in recovering many more loci at the expense of few additional false positives. However, for incomplete low-coverage assemblies the relaxed option results in too many false positives, and we recommend a safer strict option for such situations.

Assessing the performance of different search engines with ALiBaSeq, we highlight the outstanding performance of discontinuous megablast. Its forward search time on our datasets was only a few seconds, while the number of loci found was comparable with the protein-based searches and the number of false positives was smaller. We speculate that this outcome is due to the gapped initial match, which allows for variable third codon positions in between more conserved positions. Even with a hit reduced to a 9 bp initial match (default at 11 bp), blastn did not achieve the same result, as it requires an exact initial match. Due to its sensitivity and computational efficiency, dc-megablast may thus be a viable alternative to costly protein-based searches. Nucleotide-based HMMER search

(nhmmer) was generally as effective in finding loci as protein-based BLAST searches, and had less false-positives, but took several times longer to run. Protein-based HMMER search (phmmer) was about as effective as the nucleotide version but had a much higher number of false-positives.

Finally, we want to emphasize that our software has a small list of dependencies and is easy to install locally or on a bioinformatics cluster. Although a software of choice is needed to generate alignment tables, it is not required to be installed, and the actual alignment search can be done on a different machine or environment. Since the alignment software is external to the script, it can be run with a particular setting independent of our script and thus users are not constrained with default settings. Simple input and output formats allow for easy incorporation of this software as a step into existing pipelines. In addition to phylogenetic applications, the software can be used as a general similarity search results parser and sequence extractor. Even when only working with search results tables and without access to FASTA files, ALiBaSeq outputs extensive logs and query-hit correspondence tables, which include a list of contigs found for each bait, their combined score and coordinates, as well as a list of baits located on the same contigs. We provide simple easy to follow tutorials for several examples on how to use the program as well as a test dataset to guide usage.

## CONCLUSIONS

Our software is capable of retrieving orthologs from well-curated or unannotated, low or high depth shotgun, and target capture assemblies as well or better than other software as assessed by recovering the most genes with maximal coverage and with a low rate of false positives throughout all datasets. When assessing this combination of criteria, ALiBaSeq is frequently the best evaluated tool for gathering the most comprehensive and accurate phylogenetic alignments on all types of data tested. The software (implemented in Python), tutorials, and manual are freely available at https://github.com/AlexKnyshov/alibaseq.

## ACKNOWLEDGEMENTS

We would like to thank Alisa Vershinina and the Weirauch lab members for helpful comments on the manuscript. Christopher Owen and two anonymous reviewers are acknowledged for reviewing and significantly improving the manuscript.

### Funding

This work was supported by the National Science Foundation (grant number 1655769 to Christiane Weirauch). The funders had no role in study design, data collection and analysis, decision to publish, or preparation of the manuscript.

## Grant Disclosures

The following grant information was disclosed by the authors:
National Science Foundation: 1655769.

## Competing Interests

The authors declare that they have no competing interests.

## Author Contributions

- Alexander Knyshov conceived and designed the experiments, performed the experiments, analyzed the data, prepared figures and/or tables, authored or reviewed drafts of the paper, and approved the final draft.
- Eric R.L. Gordon conceived and designed the experiments, analyzed the data, authored or reviewed drafts of the paper, and approved the final draft.
- Christiane Weirauch analyzed the data, authored or reviewed drafts of the paper, and approved the final draft.

## Data Availability

The software is available for at GitHub: https://github.com/AlexKnyshov/alibaseq.

Genomic assemblies were obtained from VectorBase, assembly versions RproC3.3 and ClecH1.3:

https://vectorbase.org/vectorbase/app/downloads/Current_Release/RprolixusCDC/fasta/data/ (for RproC3.3) and https://vectorbase.org/vectorbase/app/downloads/Current_Release/ClectulariusHarlan/fasta/data/ (for ClecH1.3).

Raw data of the hybrid capture is available at NCBI SRA: SRR7819296.

## Supplemental Information

Supplemental information for this article can be found online at http://dx.doi.org/10.7717/peerj.11019#supplemental-information.

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
