# Peer review of "New alignment-based sequence extraction software (ALiBaSeq) and its utility for deep level phylogenetics"

_PeerJ, doi:10.7717/peerj.11019_

## Round 0.1 · original submission · Major Revisions

Dear Dr. Knyshov and colleagues:

Thanks for submitting your manuscript to PeerJ. I have now received three independent reviews of your work, and as you will see, the reviewers raised some concerns about the research. Despite this, these reviewers are optimistic about your work and the potential impact it will have on research studying bioinformatics tools for broad-level phylogenomics. Thus, I encourage you to revise your manuscript, accordingly, taking into account all of the concerns raised by both reviewers.

In your revision, please thoroughly and clearly illustrate how ALiBaSeq differs from existing software. Emphasize these differences in the Materials and Methods section and their discussion where they evaluate software performance. A table might be an effective means for this as well.

Overall, the relevance of your study and program would be more convincing with a robust comparison to other approaches. The reviewers provide examples. Please consider this in your revision.

Therefore, I am recommending that you revise your manuscript, accordingly, taking into account all of the issues raised by the reviewers.

I look forward to seeing your revision, and thanks again for submitting your work to PeerJ.

Good luck with your revision,

-joe

·

Basic reporting

no comment

Experimental design

Overall, the experimental design meets PeerJ standards; however, I did find it a bit difficult to determine how exactly the software differs from existing software. That being said, I am sure the software is different and I suggest the authors emphasize the differences in the Methods section and their discussion where they evaluate software performance.

Validity of the findings

no comment

Additional comments

Introduction
• Lines 35-37 ‘Because of the relative…type of sequence data’: I think this should be re-worded because it sounds like the reason orthologous genes are used is due to the ease of estimating homologs. We use orthologs because they are the result of speciation events, while paralogs are due to duplications. Using paralogs violates some assumptions of phylogenetic models and will most-likely disagree with the true species tree.
• Lines 44-45 ‘As sequencing costs…genome skimming has become…’: Is it possible to define ‘genome skimming’ and even supply a citation or two? I know what you mean here but, I just reviewed another manuscript and the author was referring to genome skimming as a method that includes using restriction enzymes to cut the gDNA (i.e., RAD approach). I am worried that ‘genome skimming’ is becoming a term used interchangeably with different methods.
• Line 51 ‘However, data obtained…’: Here, is ‘data’ referred to as the raw sequence data or the assemblies from genome skimming? I ask because on line 54 you refer to ‘annotations are unavailable’ and I think this will have to be modified if you are referring to the raw data because the annotations then fall on the scientist generating the data.
• Lines 55-57 ‘As opposed to typical…and non-target loci’: This sentence needs a citation. Also, what if the targeted locus resides in a genomic region with relatively higher G+C content?
• Lines 60-61 ‘This potential lack…currently available software.’: Why does the lack of a closely related reference make this challenging? On Line 35-36 you say, ‘…relative ease to determine homology…’.
• Lines 64-75: I like this paragraph describing these software tools. One suggestion is rephrasing the second sentence to mention that raw/curated reads are usually assembled with the help of reference orthologs. I mention this because ‘identifies reads of interest’ is a little confusing here because you are specifically targeting loci that you pre-determined as homologous/orthologous.
• Lines 76-95: I would also include OrthoFinder (Emms & Kelly 2019; https://doi.org/10.1186/s13059-019-1832-y) as an additional piece of software for this type of software
• Lines 78-79 ‘However, especially for…time-demanding procedure’: I am not sure I agree with this statement. I have used HaMStR for taxa that share a most recent common ancestor (MRCA) >300 Ma and did not notice any memory issues or time-demanding processes because of the age of the MRCA. Instead, I would argue the numbers of loci and taxa are the rate-limiting items.
• Lines 83-84 ‘However, genomic assemblies…of varying sizes.’: I am not sure I see your point here because genome assemblies include predicted proteins, which are the necessary data for many of the programs you listed here. Are you referring to generating genome skimming data and not annotating it?
• Lines 92-93 ‘Phyluce was originally…longer multiexon genes.’ I disagree with this statement. Sure, the probes sites are highly conserved, but Phyluce does not need a reference to assemble a locus; therefore, sequence conservation is irrelevant here. Furthermore, the sequences are assembled de novo using Abyss, Trinity, or Velvet, which will have no impact on ‘longer multiexon genes’ because the locus content is not considered when assembling loci.
• Line 111 ‘…no special software…’: I am not sure what you mean here. Can you clarify this?

Methods
• Line 134 ‘like BLAST and HMMER’: Can you list all of the alignment programs that your software can parse the output?
• Line 135 ‘For each query-hit pair…’: you may want to consider changing ‘hit’ to ‘subject’. I suggest this because you already use the terminology from the BLAST application (e.g., query and HSP), and using ‘Subject’ would continue the trend. BLAST is arguably the most used bioinformatics software and I think most would welcome the familiar term usage.
• Line 150 ‘…on divergent taxa.’: If the taxa share an MRCA so long ago that homology may be an issue with DNA, why not use the amino acids? Using amino acids to infer homology and orthology is standard practice in the genomics realm.
• Line 155 ‘ranked by alignment scores.’: What are the alignment scores and how are they used here? How do you distinguish between two alignments with the same score or nearly the same score? This second question is driven by using genome skimming data and the potential for multiple alleles. For example, let’s say your bait is a protein-coding exon with one site that is heterozygous. How does your software choose between the two assembled alleles?
• Lines 168-169 ‘The procedure…on the alignment tables’: Since the outcomes of your software depends heavily on alignment software, I think it is best to include the specific command lines that you use in your example below. Furthermore, based on your experience, can you offer suggestions on settings for each of your alignment programs your software uses? I think this would go a long way with your users because I am very familiar with BLAST and there are a TON of options available.
• Line 170 ‘BLAST, HMMER, LASTZ’: I think it is best to include version numbers for the software used in case the output format changes in the future. That being said, does the output from BLAST need to be in a specific format (e.g., csv, tsv, xml, etc.)? Also, does your software work with BLASTN, BLASTP, tBLASTX, etc?
• Line 184 ‘…evaluate the software’s performance…’: I think you should be specific in regard to what performance means here. For example, does performance mean computation time to perform an operation, computational efficiency (e.g., least amount of RAM needed), etc? Also, I think you should be specific in regard to your goals with comparing software. For example, are you testing the ability to assemble homologs from the sequence data and/or testing how well you can predict orthologs from the assembled homologs using RBH? One thing to keep in mind is that your RBH criteria may differ from the OrthoDB criteria for orthology prediction; therefore, your final set of predicted orthologs may differ from OrthoDB and the other programs depending on their specific criteria. I think it is important to standardize the aspect as you did with other settings from software that you describe.
• Line 190 ‘regions’: Do you mean ‘orthologs’ or ‘orthologs and adjacent non-coding regions’?
• Line 237 ‘truncated alignments until…’: Do you mean trimmed the alignment till both sequences were equal length? “both sequences were present” is throwing me off because both are present because you just aligned them with MAFFT.
• Lines 257-258 ‘The default settings…causing suboptimal performance’: Why are they inappropriate? Why not use the amino acids instead of nucleotides for the deep divergences you are trying to assemble?
• Lines 288-289 ‘We additionally capped…performance comparison fair’: Can you describe the RAM requirements, if any? Also, can ALiBaSeq be optimized on a High Performance Computing Cluster with different schedulers?
• Lines 304-305 ‘We also evaluated…assembly-based tools only).’: I agree that introns can be difficult to assemble since they typically are not part of the bait sequence(s). But the other difficulty with the introns is repetitive motifs within them. It may be a good idea to touch on this with your simulations and mention the abundance of repeats and whether or not that impacts your performance with some of these. My guess is that all of these software applications struggle with them.

Results
• Line 326 ‘…around 1,500 loci,…’: In the results section you use terms and symbols referring to uncertainty or estimates (e.g., ‘around’, ‘~’, etc.). Since they are results, can you specify the exact numbers for each?

Discussion
• General questions: What is different about your code relative to other assembly-based software that makes it slower or faster than them? Is there any particular reason(s) why researchers should use your software over the others? The latter question is just to help you pitch your software to the audience. It may be a good idea to clearly emphasize these reasons in the discussion.
• Lines 457-460 ‘With regard to…additional false positives’: I find this interesting because I am not sure why you obtained this result. It may be useful to your audience if you expand on this. Also, the ultimate question may be whether a systematist wants to relax the RBH to get more loci, but also adding potentially poor signal in their datasets by adding false positives.
• Line 475 ‘on a bioinformatics cluster’: I made a comment above regarding this, but you may want to also add something about it in the methods too.

Figures
• Figure 1: Since the requirement for your program is assembled contigs, then you may want to consider adding something to the left of ‘NCBI BLAST, HMMER…’ representing that initial step needed that your program does not perform.
• Figures 2 & 3 & 4 & 5: I can not read the left y-axis, but I am not sure if that is due to the review copy that I have.

Github Repo
The repo looks good and I looked over the code and it is well-written; however, please note that I did not test it (I can if the paper gets kicked back to me again). One thing I would suggest is to work-up a simple test dataset (e.g., 2 genes and 5 taxa) that is used with some form of a tutorial. I suggest this because, as a user, I typically choose software that has the best documentation, tutorials, and simple test data. For example, I primarily use Phyluce and Hybpiper because, in my opinion, the documentation is second to none. I would encourage you to have a look at their docs and think about possibly adding a thorough tutorial to your Github repo.

Reviewer 2 ·

Basic reporting

The paper presents a method to stitch the output from a sequence similarity program like BLAST and present homologs. The method is limited in novelty but also interesting because it can consider similarity sequence output from any program. Overall the study is fine, the simulation is okay although I think performance on more pair of genomes would be interesting to report (perhaps of varying divergence).

My main concern is the lack of recent methods that use exact alignment and high performance computing to identify similarity between divergent genomes. For example take NextGenMap, MaxSSmap, SSW, and CUDA-SW++. Each of these have shown to capture distant homologs more accurately than hash-table based searches of BLAST and HMMER. Does AliBaSeq identify more homologs if exact alignment methods were used to identify sequence similarity? The answer would be interesting either way.

Experimental design

The design is fine but I think showing performance on more pair of genomes would be interesting. A graph showing performance on increasing divergence.

What read lengths were simulated? I don't see that mentioned in the paper.

Validity of the findings

The authors conclude their method works but I'm not convinced. I think when it comes to a study claiming to find similarity at the deep phylogenetic level you have to include recent methods designed for high divergences and show more data (perhaps simulation).

Reviewer 3 ·

Basic reporting

Generally, I think that “alibaseq” is a useful tool/software for large-scale analyses of multilocus datasets, which may not be limited to protein coding genes. It is certainly of interest to many biologists. The article is well written, features and utilities are explained well, and advantages of alibaseq is highlighted compared to other software designed for similar use. But a couple of concerns:

a) The article seems better suited for a Bioinformatics Software or Methods paper than a research article.
b) Even though the article is well written overall, the writing needs to be polished in the beginning – abstract and the first few sentences of introduction. Somewhat odd grammar/structure interrupted my concentration and slowed the reading.

Addressing the following might help clarity of description.

1) False positives of what: Retrieval of orthologs?
2) Is the problem of false positives more relevant to eukaryote genome assemblies? it will be useful to specify that.
3) “Whole genome sequence’ (WGS) data” sounds simpler than ‘whole genomic sequencing’.
4) May be It can be stated that the software/pipeline is geared toward phylogenetic analyses using nucleotide sequences/WGS data.

W.r.t (3), the statement “Available tools especially struggle at deep phylogenetic levels and necessitate amino-acid space searches, increasing rates of false positive results.” in the abstract seems to contradict one of the very first results (lines 323-325):

“Protein based approaches (tblastn, tblastx, and 324 phmmer) performed best, with relaxed nucleotide-based searches (blastn with wc9 and 325 discontinuous megablast) performing almost as well.”

Experimental design

-- not applicable --

Validity of the findings

-- not applicable --

---

## Round 0.2 · Major Revisions

Dear Dr. Knyshov and colleagues:

Thanks for revising your manuscript. The reviewers are somewhat satisfied with your revision; however, there is an issue that was overlooked in the first revision. Please address these ASAP so we may move towards acceptance of your work.

Best,

-joe

·

Basic reporting

no comment

Experimental design

no comment

Validity of the findings

no comment

Additional comments

The authors have satisfactorily addressed all of my comments. Furthermore, I did not identify any other issues that need to be addressed.

Reviewer 2 ·

Basic reporting

No comment

Experimental design

No comment

Validity of the findings

No comment

Additional comments

I appreciate the author rebuttal comments but I believe they are slightly flawed. All three tools NextGenMap, SSW, and MaxSSmap actually produce HSP regions which is why I suggested them in the first place. Yes, both BLAST and HMMER are far more widely used and from that perspective it makes sense to have ALiBaSeq work for them. But both are also weak on divergent data. I'm not sure if a claim on utility for deep-level phylogenetics makes sense if baseline methods designed for high divergences are ignored. This claim is a main motivator of ALiBaSeq as given in the second sentence of the abstract "Available tools especially struggle at deep phylogenetic levels and necessitate amino-acid space searches, which may increase rates of false positive results."

This can easily be addressed by evaluating on a pair of more divergence species. The result would likely show missed regions and a suggestion for using baseline tools for higher divergence may be made (perhaps as part of future work).

---

## Round 0.3 · accepted · Accept

Dear Dr. Knyshov and colleagues:

Thanks for revising your manuscript based on the concerns raised by the reviewers. I now believe that your manuscript is suitable for publication. Congratulations! I look forward to seeing this work in print, and I anticipate it being an important resource for groups studying bioinformatics tools for broad-level phylogenomics. Thanks again for choosing PeerJ to publish such important work.

Best,

-joe